# Antioxidant Activity and Sensory Quality of Bacon

**DOI:** 10.3390/foods11020236

**Published:** 2022-01-17

**Authors:** Bing Zhou, Jie Luo, Wei Quan, Aihua Lou, Qingwu Shen

**Affiliations:** College of Food Science and Technology, Hunan Agricultural University, Changsha 410128, China; zb11020711@163.com (B.Z.); 7170112074@stu.jiangnan.edu.cn (J.L.); reus_quan@hunau.edu.cn (W.Q.); louaihua916@163.com (A.L.)

**Keywords:** bacon, liquid smoke, antioxidant capacity, protein oxidation

## Abstract

Effects of liquid smoke prepared from different woods on physicochemical parameters, sensory quality, and protein and lipid oxidation were determined in bacons during process and storage. The relationship between the antioxidant activity of smoked liquid and the quality of bacon was further explored through chemometric analysis. Results showed that liquid smoke prepared from different woods differed in phenolic and carboxyl compounds and antioxidant capacity. Bacon processed with different liquid smoke had different antioxidant capacity, lipid and protein oxidation during storage, and sensory quality. The concentration of phenols was positively highly correlated with the antioxidant capacity of both liquid smoke and fresh bacon, but negatively correlated with lipid and protein oxidation in bacon. Among the five woods, liquid smoke made from *Punica granatum* L. showed higher antioxidant capacity, but bacon smoked with *Armeniaca vulgaris* Lam had better overall eating quality. This study reveals that selection of woods to prepare antioxidant fumigant is a feasible approach to retard oxidative spoilage of meat products. Future study is need for the development of composite smoke flavorings to improve both oxidative stability and sensory quality of foods.

## 1. Introduction

Bacon is a processed meat product made from a pig belly or back that is subjected to curing, drying, and smoking. Due to its unique smoked flavor, bacon is widely consumed in western countries and even in the world [1], while the consumption in China was low in the past though historically the first bacon might be made in China [2]. In recent years, bacon and some other western style meat products are becoming increasingly popular in China, especially among young people with the change of eating habits and lifestyle. Although Maillard reaction and lipid oxidation contribute to the generation of some volatile compounds in it, smoking is the predominant source of volatile compounds and mostly influence bacon flavor [3,4]. For this reason, smoking agents have to be developed to meet the taste preference of local people in different region of the world.

Bacon is rich in lipids and proteins, making it prone to oxidation. Oxidation of lipids, especially poly-unsaturated fatty acids, entails the generation of rancid or off flavor, decreases the nutritional value and reduces the storage period of foods [5]. In addition to lipid oxidation, protein oxidation has recently attracted the attention of food scientists due to its detrimental effect on food quality and human health. First, protein oxidation alters not only the eating quality of fresh meat (e.g., meat tenderness and juiciness), but also the processing properties of meat (e.g., solubility and gelation) [6]. Second, oxidative modification of proteins can lead to loss of essential amino acids and reduces protein digestibility, and thus the nutritional quality of foods [7,8]. Third, some products from dietary protein oxidation are toxic and have been reported to promote inflammatory conditions, linking to the onset of carcinogenesis in the gut. In addition, some oxidative species from dietary proteins interfere with cell metabolism, alter gene expression and induce a variety of health disorders or diseases after uptake into cells [9]. Finally, oxidation of myoglobin to metmyoglobin causes discoloration of meat and meat products. Some studies reported that antioxidants, such as vitamin E and some other plant extracts could be used to reduce lipid and protein oxidation induced deterioration of foods [10,11]. For this reason, smoking agents with antioxidant activity have become a choice to improve the oxidative stability and shelf life of smoked meat products in recent years [5,12].

In addition to flavor and oxidative stability, the safety of bacon products is also a very important issue [13]. Compared with the traditional smoking process in which meat is exposed to the smoke from burned sawdust or woodchips, the use of liquid smoke is being increasingly used as an alternative with many advantages, such as elimination of carcinogenic compounds (e.g., polycyclic aromatic hydrocarbons), reduced processing time, lower environmental pollution, and less smoke varieties [14,15,16]. In general, all of the major chemical compounds of wood smoke, whether using a traditional or a liquid smoke condensate approach, can be grouped into the following groups: acid compounds, phenolic compounds, and carbonyl compounds [17]. Hardwoods consist of three main materials: cellulose (50%), hemicelluloses (25%), and lignin (25%). Cellulose and hemicelluloses are aggregate sugar molecules and the pyrolysis of cellulose and hemicelluloses forms significant amount of carbonyls, which is responsible for the primary brown color on the surface of smoked meats. Lignin is a complex arrangement of interlocked phenolic molecules, while the pyrolysis of lignin forms some phenolics which give the desirable flavor and act as antioxidative compounds for smoked meat [17,18]. Common wood types include hardwoods, fruitwoods, and nut woods which can be used for smoking meat. However, present studies revealed that it is the differences that exist in these compositions within different wood (based on tree species) that account for the quality differences observed in smoked meat products with respect to color, flavor, preservation, shelf-life, etc. [18,19,20].

Therefore, in the present study, liquid smoke was prepared from different woods (including hardwoods, fruitwoods, and nut woods) and implemented on bacon. The antioxidant capacity of liquid smoke, oxidative stability and sensory quality of bacon, as well as volatile compounds in both liquid smoke and processed bacon, were comparatively evaluated with the aims for high quality bacon production and further understanding of the factors influencing bacon shelf life and wholesomeness.

## 2. Materials and Methods

### 2.1. Preparation of Liquid Smoke

Liquid smoke was prepared by dry distillation as reported in literature [21] with slight modification instead. Briefly, 200 g of wood were put into a dry distillation bottle and heated to 680 °C. The dry distillation device was mainly composed of an electric heating sleeve with automatic temperature control, a serpentine condenser tube and a fraction collection device. The flue gas, after 15 min of smoking, was collected through a serpentine condenser device. The obtained condensate was purified by precipitation, stratification, filtration through double-layer filter paper, and absorption with food-grade cylindrical-granular-activated carbon. Five different liquid smokes were prepared from five different woods, *Cupressus funebris* Endl (CE), *Armeniaca vulgaris* Lam (AL), *Diospyros kaki* Thunb (DT), *Punica granatum* L. (PL), and *Ziziphus jujuba* Mill (ZM). For each type of wood, liquid smoke was made in six replicates. A commercial liquid smoke (CS) was purchased from Red Arrow International LLC (Manitowoc, WI, USA) and used as a reference in the study.

### 2.2. Bacon Preparation

The cuts of bacon (pig tenderloin) were derived from a carcass of crossbreed pig (Duroc × Landrace × Yorkshire). The pork samples were cut into pieces (4 cm × 4 cm × 1.5 cm) before being cured/smoked. For bacon antioxidant capacity and oxidative stability analysis, pork cuts were submerged in curing/smoking solution (5.5% salt, 0.1% liquid smoke) and cured for 12 h at 4 °C. Afterwards, samples were cooked in a baking oven at 70 °C for 4 h, individually wrapped in plastic water proof bags and stored at 4 °C for 9 days. On day 0, 3, 6, and 9 the bacon samples were collected and stored at −80 °C until analysis. For sensory evaluation and volatile compound analysis, pork cuts were cured in curing/smoking solution (5.5% salt, 1% liquid smoke, 0.5% sodium nitrite and 0.1% ascorbic acid) for 12 h at 4 °C and cooked in a baking oven at 70 °C for 4 h before analysis.

### 2.3. Physico-Chemical Analysis

The total phenol content in liquid smoke was measured by Folin–Ciocalteu method as in literature [22] with some modification instead. Briefly, 200 µL of diluted (500×) liquid smoke were added with 1 mL of Folin–Ciocalteu phenol reagent (Sigma-Aldrich, St. Louis, MO, USA) and vortexed thoroughly. Then, 800 µL of 7.5% Na_2_CO_3_ were added and incubated at 25 °C in the dark for 1 h. The absorbance at 765 nm was determined with a Varioskan Flash microplate reader (Thermo Scientific, Rockford, IL, USA). Distilled water was used for blank. Gallic acid was used to prepare the standard curve and total phenol in samples was calculated as gallic acid equivalents.

The total carbonyl compounds in liquid smoke were determined according to GB1886.127-2016 National Food Safety Standards–Food Additive Smoke Flavor I and II from Haw-pit. Carbonyl compounds reacted with 2,4-Dinitrophenylhydrazine (2,4-DNPH) in acidic condition to generate red hydrazone derivatives, which were determined by measuring the absorbance at 430 nm. Total carbonyl compounds in liquid smoke were calculated as heptaldehyde equivalents.

Benzo (a) pyrene (BaP) in liquid smoke were determined according to GB5009.27–2016 National Food Safety Standards–Determination of Benzo (a) pyrene in Foods. BaP in meat samples was extracted by n-hexane, purified by BaP imprinted column, and determined by a Waters 1525 HPLC coupled with fluorescence detector and Symmetry C18 Column (100 Å, 5 µm, 4.6 mm × 250 mm, 1/pk, Waters company, Milford, MA, USA). Isocratic elution mode was selected, with a mobile phase consisting of 88% acetonitrile-water (acetonitrile: ultrapure water, 88:12, *v*:*v*) at a flow rate of 1.0 mL/min. The excitation and emission wavelength of fluorescence detector was 384 and 406 nm, respectively, the column temperature was 35 °C, and the injection volume for each analysis was 20 μL.

### 2.4. Antioxidant Activity Assay

The antioxidant activities of liquid smoke and bacon were evaluated by DPPH (2,2-Diphenyl-1-picrylhydrazyl) free radical scavenging capacity, ABTS^+^ (2′-azino-bis [3-ethylbenzthiazoline-6-sulphonic acid]) scavenging capacity, total antioxidant capacity (T-AOC), and oxygen radical absorbance capacity (ORAC). Liquid smoke was diluted before the assay. Then, 0.1 g bacon samples were homogenized in 1 mL of precooled PBS (phosphate buffered saline) and centrifuged at 4 °C and 10,000× *g* for 5 min. The supernatant was used for antioxidant activity assay.

DPPH assay was performed as previously described [23]. DPPH was dissolved in methanol at a final concentration of 0.3 mM. Then, 50 μL of the sample, 100 μL of 95% ethanol, and 50 μL of DPPH solution were added to a well in a 96-well plate. After incubation at 25 °C for 30 min, the absorbance at 515 nm was measured. The DPPH radical scavenging rate was calculated as in literature [23].

ABTS assay was conducted as described by Roberta et al. [24]. Briefly, ABST radical cation (ABTS•+) was produced by reacting 7.0 mM ABTS stock solution with 2.45 mM potassium persulfate in dark at 25 °C for 12–16 h. The ABTS•+ solution was diluted to A_734_ = 0.700 with PBS to make ABTS•+ working solution. Then, 10 µL of diluted liquid smoke or bacon extract were added with 200 µL of ABTS•+ working solution, mixed well and reacted at 30 °C for 10 min. The absorbance at 734 nm was measured. Ultrapure water was used for a blank measurement. The ABTS free radical scavenging rate was calculated to evaluate ABTS free radical scavenging ability of samples [24].

The T-AOC of liquid smoke and bacon was measured by using a T-AOC Assay Kit (Solarbio, Beijing, China). The ORAC values of samples were determined according to literature [25]. Briefly, 25 μL of sample or calibration standard Trolox (0–100 µM) were loaded on to a 96-well plate. Then, 150 µL of 0.035 µg/mL fluorescein solution was added. After equilibration, 25 µL of a solution of 2,2′-azobis (2-methylpropionamidine) dihydrochloride (AAPH) were added and fluorescence was measured [25]. ORAC was calculated as Trolox equivalents.

### 2.5. Protein and Lipid Oxidation Analyses

Carbonyl content, sulfhydryl and disulfide bond content, protein surface hydrophobicity (PSH), and 2-thiobarbituric acid reactive substance (TBARS) values of meat were measured using our previous protocols [26]. Meat fluorescence was measured as in literature [27].

### 2.6. Sensory Evaluation

The sensory evaluation was conducted using a 10-person trained taste panel. The panelists were selected and trained according to literature [28]. Before sensory evaluation, bacon slices were cooked in a baking oven at 160 °C for 5 min. The evaluation was then finished in a single session as previously reported [19]. Briefly, samples were present in disposable cups to panelists in individual booths. Water and bread were used to cleanse the palate between samples. All samples were scored on a 1–10 point scale for bacon appearance (shine, redness, and yellowness), flavor (saltiness and smokiness) and texture (fattiness, succulence, and chewiness) as in literature [28]. At the same time, testing samples were scored in comparison to a commercial available bacon for meaty odor, pleasant odor, unpleasant odor, pleasant taste, and overall liking as previously described [29].

### 2.7. Statistical Analysis

The results reported are the means of six replicates (expressed as the mean ± standard error). The data from various treatments were analyzed by one-way analysis of variance (ANOVA) by using SPSS software package version 25.0 (SPSS Inc., Chicago, IL, USA). Levels for significant differences were set at *p* < 0.05. PCA and correlation analysis were performed using OriginPro 2021 (OriginLab Corporation, Northampton, MA, USA).

## 3. Results and Discussion

### 3.1. Total Phenols, Carbonyl Compounds, and BaP in Liquid Smoke

From the point of view of chemical composition, both smoke and smoke flavorings are very complex mixtures, which depend on the type of wood, the pyrolysis conditions and the treatments of the smoke. According to literature, the smoke produced at 650–700 °C is richest in components able to impart desirable organoleptic properties to treated products [30]. Thus, a temperature of 680 °C was selected for carbonization in the present study. The composition of smoke has been extensively studied in recent years and more than 2000 compounds were identified. These compounds belong to many different chemical classes, including aldehydes, ketones, alcohols, acids, esters, furan and pyran derivatives, phenolic derivatives, hydrocarbons, and nitrogen compounds. Among them, the phenolic fraction probably represents the most important one both from the qualitative and quantitative point of view as phenolic compounds are closely related to the fumigation flavor and the antioxidant activity of smoke flavorings. Previous studies reveal that most phenolic compounds have fumigation flavor [31,32] and the antioxidant activity of liquid smoke is well correlated with the concentrations of phenolic compounds [33]. For these reasons, the total phenols in liquid smoke were measured by Folin–Ciocalteu method in the present study. As shown in Figure 1A, the liquid smoke made from PL had the highest content of phenols, which reached 29.33 ± 0.85 mg/mL, followed by the smokes prepared from AL, ZM, CE, and CS. In addition, the concentrations of total phenols in all liquid smokes prepared in lab were higher (*p* < 0.05) than that of the commercial liquid smoke (CS). The higher concentration of total phenols indicates that the liquid smoke prepared from PL may be superior in fumigation flavor and antioxidant activity.

Carbonyl compounds represent the largest number of compounds in wood smoke [34]. This class of compounds contributes to the overall sensory properties of wood smoke. It is reported that some compounds of this class posses a caramel or burnt sugar aroma that modifies the flavor of phenols [35]. In addition, carbonyl compounds provide most of the colour components and their content is closely related to the colour of fumigant [21,36]. For these reasons, the carbonyls in prepared liquid smoke were measured in the present study. As shown in Figure 1B, liquid smoke prepared from CE and PL had significantly higher (*p* < 0.05) carbonyl compounds than other liquid smokes. As most carbonyls are formed from the thermal decomposition of cellulose and hemicelluloses through carbohydrate degradation, The difference in carbonyl content should be related to the variation in chemical composition of different woods [30]. Based on the content of total phenols and carbonyls, it seems that the wood of PL had advantage for liquid smoke preparation.

Polycyclic aromatic hydrocarbons (PAHs) are a group of compounds which are harmful to human health. PAHs are formed in the decomposition of wood, especially at limited access of oxygen in the range of 500–900 °C [30]. Benzo[a]pyrene (BaP) is accepted as the indicator of total PAHs in smoked foods [30]. BaP is carcinogenic and limitation in foods has been legislated in different counties. HPLC analysis showed that no detectable BaP was determined in the present study, showing that the purification process was effective to eliminate BaP in prepared liquid smoke.

### 3.2. Antioxidant Activity of Liquid Smoke and Bacon

Different assays have been introduced to measure antioxidant capacity of foods and biological samples, which describes the ability of redox molecules in foods and biological systems to scavenge free radicals. Antioxidant capacity assay measures the additive and synergistic effects of all antioxidants rather than the effect of a single compound; therefore, it may be useful to evaluate the potential benefits of antioxidants on food nutrition, quality, and safety [5,37]. In addition, oxidation of myoglobin to metmyoglobin causes discoloration, shortening the shelf life of meat and meat products. For these reasons, increasing attention has been paid to the antioxidant capacity of liquid smoke [12].

DPPH assay showed that liquid smoke prepared from different woods had different DPPH free radical scavenging capacity, which increased with the decrease in dilution (Figure 2A). For all dilutions, liquid smoke made from PL and AL generally showed higher (*p* < 0.05) DPPH free radical scavenging capacity before reaching the upper limit of the measurement, which is in accordance with the higher concentration of total phenols in them (Figure 1A). In addition, all liquid smokes showed higher DPPH free radical scavenging capacity than the positive control vitamin C used in the study. In agreement, bacon processed with LP and AL smoke showed higher (*p* < 0.05) DPPH free radical scavenging capacity (Figure 2E), showing that liquid smoke could be used to modify the antioxidant capacity/oxidative stability of processed meat products. ABTS assay also showed the different ABTS^+^ free radical scavenging capacity of liquid smokes made from different woods (Figure 2B) and bacon prepared with PL liquid smoke had the highest (*p* < 0.05) ABTS^+^ free radical scavenging capacity, followed by bacon made with AL liquid smoke (Figure 2F). At the same dilution, the ABTS^+^ free radical scavenging rates were higher than DPPH free radical scavenging rates for all liquid smokes, which was consistent with literature [37]. The higher antioxidant capacity of liquid smoke determined by ABTS assay than DPPH assay could be explained by the high concentration of phenolic compounds (higher than other types of antioxidants) in liquid smoke as the antioxidant capacity detected by the ABTS assay is mainly associated with the content of phenolic compounds and flavonoids present in the sample, while the DPPH assay mainly reflects the presence of high-pigmented and hydrophilic antioxidants [37,38].

The T-AOC assay also showed that liquid smoke made from PL had higher (*p* < 0.05) and the commercial liquid smoke had lower (*p* < 0.05) antioxidant capacity when compared with others (Figure 2C). Similarly, liquid smoke made from PL showed higher (*p* < 0.05) ORAC value than CE, AL, DT, and the commercial liquid smoke (Figure 2D). Consequently, the bacon prepared with PL liquid smoke showed higher (*p* < 0.05) total antioxidant capacity (Figure 2G) and oxygen radical absorbance capacity (Figure 2H), which agrees with DPPH and ABTS assays. In summary, these data reveal that liquid smokes made from different woods had different antioxidant capacity, which may contribute to the difference in antioxidant capacity and likely oxidative stability of processed bacon. Among the five different liquid smokes prepared in the study, the one made from PL showed higher concentration of phenols and higher antioxidant capacity.

### 3.3. Impact of Liquid Smoke on Lipid and Protein Oxidation in Bacon

Meat and meat products are rich in lipid and proteins, which are susceptible to oxidation during processing and storage. According to reports, oxidation of lipid and proteins is the second-most important cause of meat spoilage following microbial spoilage, which together lead to the spoilage and waste of millions of tons of food/meat during storage every year [12,39]. Oxidation causes quality degradation, reduces nutritional value, and deteriorates palatability and other organoleptic traits of meat products [6,40]. In addition, some compounds formed from lipid and protein oxidation are cytotoxic, mutagenic and oxidative to human body, thereby inducing cancer, atherosclerosis, inflammation, and other disorders or diseases [9,41]. For these reasons, plant-derived natural antioxidants and their application in meat and meat products have been extensively studied in past years [42].

The impact of liquid smoke on lipid oxidation in bacon during storage was evaluated by TBARS analysis. The value of TBARS reflects the amount of secondary oxidation products of lipid, which could generate off-flavor in meat and meat products. As shown in Figure 3A, the control unsmoked cured meat (CM, processed in the same procedure without adding liquid smoke) had significantly higher (*p* < 0.05) TBARS values than those of the smoked bacon during the whole storage period, showing that liquid smoke was antioxidative and retarded lipid oxidation in bacon during storage. When the six smoked bacon samples were compared, bacon prepared with the commercial liquid smoke showed higher (*p* < 0.05) TBARS values than the others. In addition, The lowest TBARS values were determined in the samples of the AL and PL group during the first three days of storage. This is generally in agreement with the content of phenolic compounds in liquid smoke (Figure 1A) and the antioxidant capacity of both liquid smoke and bacon (Figure 2), indicating that phenols are the major antioxidants in liquid smoke, protecting lipids from oxidation in bacon. Notably, the TBARS values that were determined on day 9 tended to decreased compared to those on day 6. This is probably because malondialdehyde (MDA) reacts with a large scale of compounds or it forms MDA dienes or trienes, resulting in a quantitative decrease of MDA available for reaction with thiobarbituric acid and thus the TBARS values [5].

The formation of carbonyl compounds is one of the most remarkable modifications in oxidized proteins and protein carbonyl content is commonly measured for the evaluation of protein oxidation [6,43,44]. Total sulfhydryl includes free sulfhydryl group exposed on the surface of the protein and sulfhydryl group embedded in the molecule. Oxidation of sulfhydryl group to disulfide bond forms intra- and intermolecular cross-linking, which may alter protein spatial structure and function. As oxidation progresses, the level of sulfhydryl reduces and disulfide bond forms. Thus, measurement of thiol content is a compatible indicator of protein oxidation [45].

The influence of smoking on protein oxidation in bacon during storage is shown in Figure 3B–D. As expected, with the increase of storage time, carbonyl content tended to increase in meat, showing the increasing protein oxidation (Figure 3B). In addition, smoking obviously inhibited protein oxidation in meat as the carbonyl content was significantly higher (*p* < 0.05) in the unsmoked cured meat throughout the whole storage period when compared to that of the smoked bacon, especially smoking with AL and PL woods reduced carbonyl content ~35.6% compared to the CM treatment. In agreement with carbonyl content in meat, the sulfhydryl content was significantly lower (*p* < 0.05) and disulfide bond was significantly higher (*p* < 0.05) in the unsmoked cured meat (Figure 3C,D), showing that smoking inhibited sulfhydryl oxidation to form disulfide bond in bacon during storage. Especially, the bacon made with AL and LP liquid smoke had significantly higher (*p* < 0.05) total sulfhydryl content on day 3, 6, and 9, and lower (*p* < 0.05) disulfide bond on day 6 compared to all the other groups, further demonstrating that the higher content of phenolic compounds (Figure 1A) and higher antioxidant capacity of liquid smoke (Figure 2A–D) are associated with lower protein oxidation in processed meat products.

Hydrophobic interaction between non-polar amino acid residue side chains is the main reason for protein folding into the tertiary structure. Oxidation of protein sulfhydryl groups alters protein conformation and the surface hydrophobicity increases [46]. From Figure 3E, it can be told that surface hydrophobicity of proteins in meat tended to increase during the whole period of storage. On day 3, the unsmoked cured meat had higher (*p* < 0.05) surface hydrophobicity than all the smoked bacon, indicating a higher degree of protein oxidation in CM and liquid smoke inhibited protein oxidation in bacon. Again, the lower protein surface hydrophobicity in AL and PL groups compared to the DT and CS groups confirmed liquid smoke prepared from AL and PL woods had higher antioxidant capacity (Figure 2A–D) and inhibited sulfhydryl oxidation (Figure 3C).

Previous studies have shown that meat fluorescence emission, which is emitted from proteins modified by aldehydes from lipid peroxidation to form Schiff bases, can be used as a marker of oxidation in meat induced by cooking [27,47]. Just as in literature, two emission peaks were seen for meat extract after excitation at 360 nm, a major peak at around 420 nm, and a minor peak at around 515 nm (Figure 3F). The height of the peak at 420 nm of the unsmoked cured meat was much greater than those of the smoked bacon, indicating the increased oxidation of this group and that liquid smoke inhibited oxidation in bacon right after cooking (Figure 3F). Noticeably, the heights of both peaks tended to decrease during storage and the peak at 515 nm almost disappeared on day 9 (Figure 3F–I), which might be attributed to the instability of Schiff bases induced by cooking.

### 3.4. Sensory Attributes of Bacon

The sensory panel evaluation of prepared bacon is shown in Appendix A. Liquid smoke prepared from different woods had significant effects (*p* < 0.05) on bacon appearance (redness and yellowness) and flavor (smokiness, meaty odor, pleasant odor, unpleasant odor, and pleasant taste). The scores of saltiness and texture attributes (fattiness, succulence, and chewiness) were not different between treatments, showing that liquid smoke prepared from different woods had no significant effect on these sensory quality attributes. The significantly higher overall liking score of AL bacon suggests that bacon smoked with AL wood received sensory panelists’ preference over bacon smoked with CE, DT, and ZM woods. To better explore the impact of smoking woods on bacon eating quality, principle component analysis (PCA) was performed. As shown in Figure 4, PC1 explained 52% of the variance associated with bacon sensory quality traits and could be explained to represent the “overall eating quality” of bacon. AL treatment located far from the rest treatments on PC1, suggesting that bacon smoked with AL wood had higher overall eating quality than others. The pleasant taste, pleasant odor, and smokiness located together with AL treatment in the fourth quadrant, while the unpleasant odor located in the second quadrant close to the CE and ZM treatment with low overall eating quality, suggesting that these flavor traits were important to the overall eating quality of bacon. The difference in overall eating quality of bacon smoked with different woods demonstrates that smoking woods had a obvious impact on sensory quality of processed bacon [3].

### 3.5. Correlation Analysis of Smoke Antioxidant Capacity and Oxidation in Bacon

To better explore the influence of liquid smoke on oxidation in processed bacon, a correlation analysis was performed. As shown in Figure 5, the antioxidant capacity of liquid smoke was positively highly correlated with the antioxidant capacity of bacon as the DPPH and ABTS+ free radical scavenging capacity, T-AOC, and ORAC values of liquid smoke were positively highly correlated with those of freshly prepared bacon (0.78 ≤ *r* ≤ 0.91, *p* < 0.001), showing that smoke should be a major resource of antioxidant compounds in processed meat products. TBARS values measure lipid oxidation in meat. The TBARS values of bacon determined on day 9 were negatively correlated with the DPPH and ABTS+ free radical scavenging capacity, T-AOC and ORAC values of fresh bacon (−0.69 ≤ *r* ≤ −0.57, *p* < 0.05). This is logic because antioxidants in bacon protect lipid from oxidation during storage. The antioxidant capacity of liquid smoke measured by DPPH, ABST, T-AOC, and ORAC assays was negatively correlated with the TBARS values of bacon on day 9 (−0.71 ≤ *r* ≤ −0.47, *p* < 0.05), especially the DPPH free radical scavenging capacity of liquid smoke was negatively highly correlated with TBARS values of bacon on day 9 (*r* = −0.71, *p* < 0.001), demonstrating that antioxidants from smoke play a role in inhibiting oxidation in bacon. This is confirmed by the negative correlation between the antioxidant capacity (DPPH and ABTS+ free radical scavenging capacity, and T-AOC values) of liquid smoke and carbonyl content in bacon on day 9 (−0.70 ≤ *r* ≤ −0.63, *p* < 0.05), the positive high correlation between the antioxidant capacity of liquid smoke and sulfhydryl content in bacon on day 9 (0.71 ≤ *r* ≤ 0.83, *p* < 0.001), the negative correlation between the antioxidant capacity of liquid smoke and disulfide bond content in bacon on day 9 (−0.74 ≤ *r* ≤ −0.51, *p* < 0.05), and the negative correlation between the antioxidant capacity (DPPH and ABTS+ free radical scavenging capacity, and T-AOC values) of liquid smoke and surface hydrophobicity of proteins in bacon on day 9 (−0.73 ≤ *r* ≤ −0.53, *p* < 0.05), which shows that smoking inhibited protein oxidation in bacon during storage. In consistence with literature [33], the present study showed that phenolic compounds in liquid smoke was positively correlated with the antioxidant capacity, especially positively highly correlated with the ABST+ free radical scavenging capacity (*r* = 0.96, *p* < 0.001) and total antioxidant capacity (*r* = 0.87, *p* < 0.001) of liquid smoke. In addition, the content of phenolic compounds in liquid smoke was positively highly correlated with the antioxidant capacity of bacon (0.62 ≤ *r* ≤ 0.86, *p* < 0.01), but negatively correlated with protein oxidation in bacon on day 9. These data suggest that phenols should be the major antioxidant compounds in smoke. Smoking significantly inhibited lipid and protein oxidation in meat during storage (Figure 3) and the selection of woods can be an approach to improve oxidative stability and shelf life of smoked meat products.

## 4. Conclusions

Liquid smoke made from different woods varies in chemical composition and antioxidant capacity. Smoking modulates the antioxidant capacity and the oxidation of lipids and proteins of bacon. This study reveals that the selection of smoking woods and development of liquid smoke with antioxidant activity are a feasible approach to enhance the oxidative stability and shelf life of produced meat products. Phenols are a major class of antioxidant compounds to modulate the antioxidant capacity of both liquid smoke and fresh bacon. Among the five types of woods studied here, liquid smoke made from *Punica granatum L* showed highest antioxidant capacity followed by liquid smoke prepared from *Armeniaca vulgaris* Lam, but bacon smoked with *Armeniaca vulgaris* Lam showed better eating quality. Future study of composite smoke flavorings (smoke flavorings made from more than one type of wood) may help to overcome the flaws of smoke liquid made from single type of wood.

## Figures and Tables

**Figure 1 foods-11-00236-f001:**
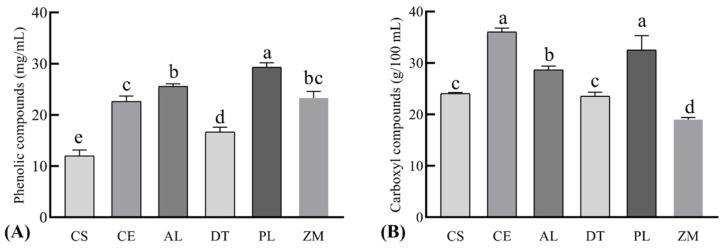
The concentrations of phenolic (**A**) and carbonyl compounds (**B**) determined in prepared liquid smoke from five different woods. CS, commercial liquid smoke; CE, *Cupressus funebris* Endl; AL, *Armeniaca vulgaris* Lam; DT, *Diospyros kaki* Thunb; PL, *Punica granatum* L.; ZM, *Ziziphus jujuba* Mill. Bars lacking common letters differ at *p* < 0.05, *n* = 6.

**Figure 2 foods-11-00236-f002:**
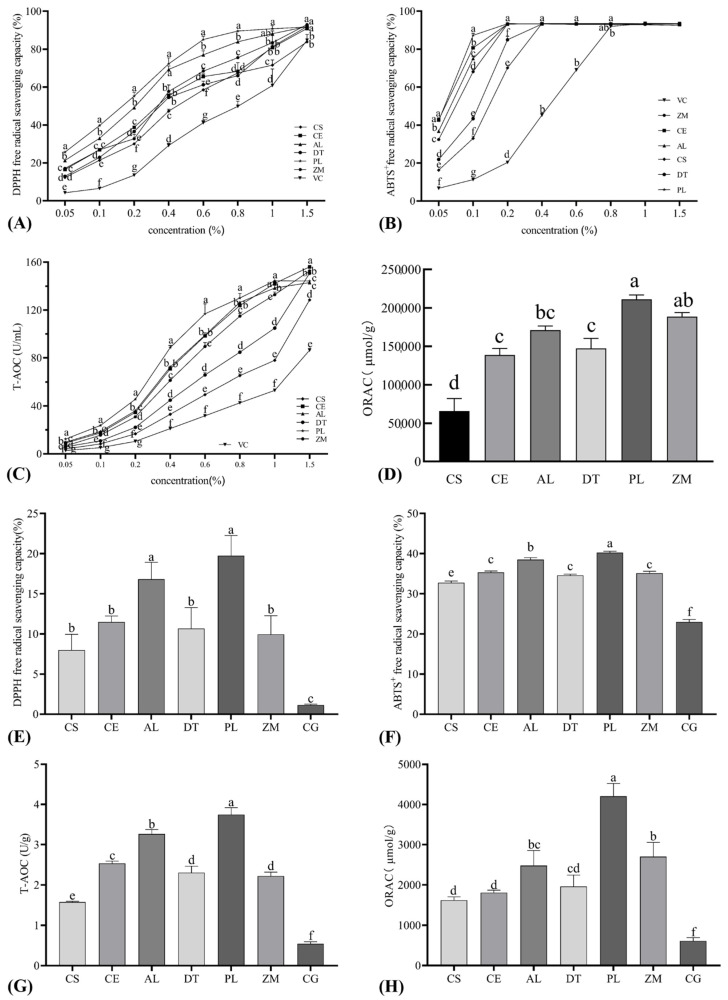
The antioxidant capacity of liquid smoke prepared from five different woods (**A**–**D**) and bacon prepared from different liquid smoke (**E**–**H**). CS, commercial liquid smoke; CE, *Cupressus funebris* Endl; AL, *Armeniaca vulgaris* Lam; DT, *Diospyros kaki* Thunb; PL, *Punica granatum* L.; ZM, *Ziziphus jujuba* Mill; VC, vitamin C; CM, unsmoked cured meat. Means at the same concentrations (**A**–**C**) and bars (**D**–**H**) lacking common letters differ at *p* < 0.05, *n* = 6.

**Figure 3 foods-11-00236-f003:**
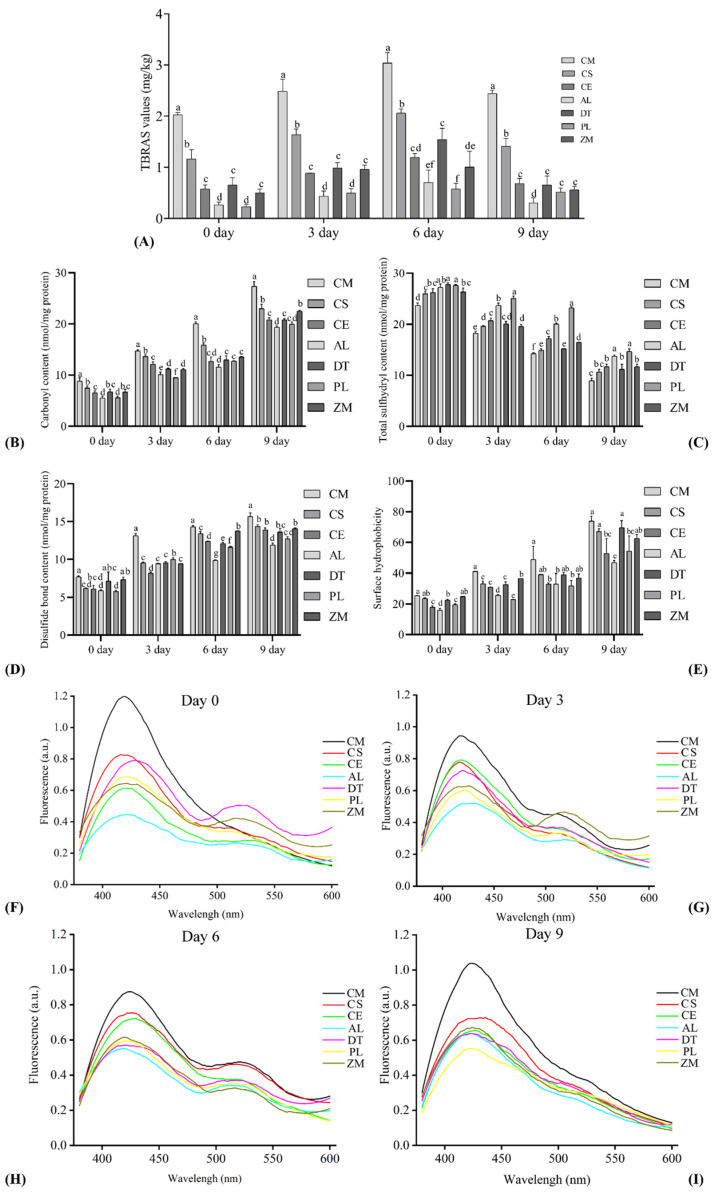
The impact of liquid smoke on lipid and protein oxidation in bacon during storage evaluated by (**A**) TBARS value (**B**) protein carbonyl content (**C**) total sulfhydryl content (**D**) disulfide bond content (**E**) surface hydrophobicity, and (**F**–**I**) meat fluorescence. CM, unsmoked cured meat; CS, commercial liquid smoke; CE, *Cupressus funebris* Endl; AL, *Armeniaca vulgaris* Lam; DT, *Diospyros kaki* Thunb; PL, *Punica granatum* L.,; ZM, *Ziziphus jujuba* Mill. Means at the same concentrations and bars lacking common letters differ at *p* < 0.05, *n* = 6.

**Figure 4 foods-11-00236-f004:**
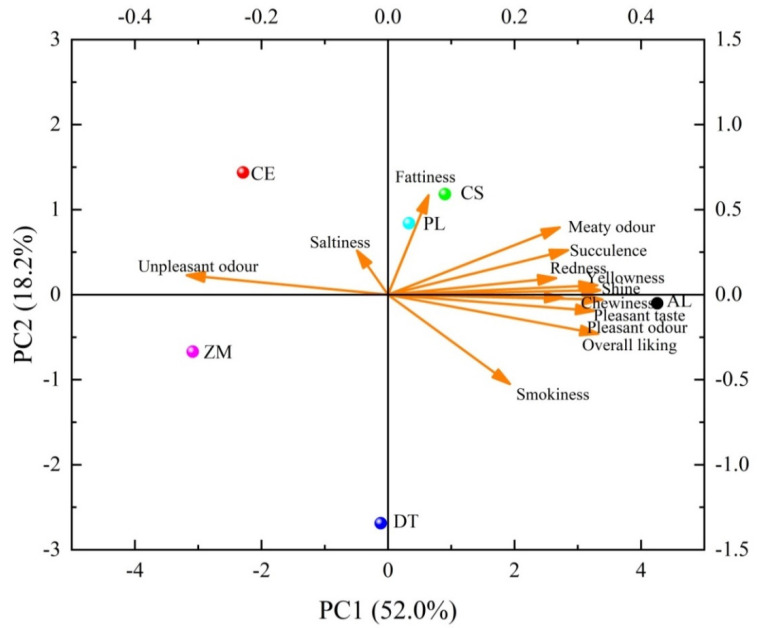
Principle component analysis (PCA) for sensory attributes of bacon smoked with different woods. CS, commercial liquid smoke; CE, *Cupressus funebris* Endl; AL, *Armeniaca vulgaris* Lam; DT, *Diospyros kaki* Thunb; PL, *Punica granatum* L.; ZM, *Ziziphus jujuba* Mill.

**Figure 5 foods-11-00236-f005:**
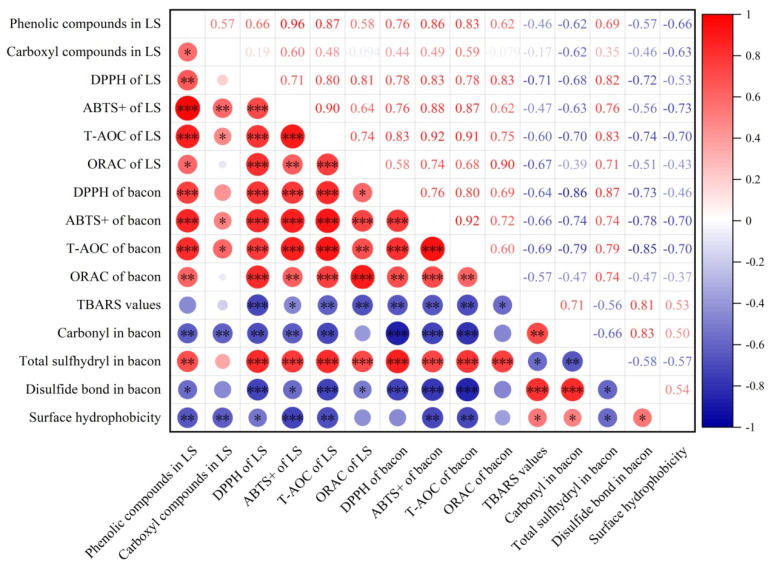
The correlation analysis of smoke antioxidant capacity and oxidation in bacon. The data of phenolic and carbonyl compounds in liquid smoke, the antioxidant capacity of both liquid smoke and freshly prepared bacon (day 0), and oxidation of lipids and proteins on day 9 were used for correlation analysis. Red represents positive correlation. Blue represents negative correlation. LS, liquid smoke. * *p* < 0.05; ** *p* < 0.01; *** *p* < 0.001.

## Data Availability

The datasets generated for this study are available on request to the corresponding author.

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
