# Peer review of "Antioxidant Activity and Sensory Quality of Bacon"

_foods, 2022, doi:10.3390/foods11020236_

Round 1
Reviewer 1 Report
Shen et al measured the antioxidant activity of smoked liquid and applied it in bacon, the point is hot and could be published, however the authors MUST fix several issues before moving to the next step.
- However, the aim of this study is to compare among different woods to produce antioxidative smoked-liquid, but there are many studies around this point, I show you some example:
- Composition of Phenolic Compounds and Antioxidant Activity of Commercial Aqueous Smoke Flavorings
- Comparative evaluation of the antioxidant capacity of smoke flavouring phenols by crocin bleaching inhibition, DPPH radical scavenging and oxidation potential
- Effects of different smoking methods on sensory properties, free amino acids and volatile compounds in bacon
So, the novelty of this study is somewhat not clear, and the authors ought to highlight what are the new things they deliver to the scientific community, why we have to read your paper and ignore others, plz consider this point in detail.
- Authors are also encouraged to ask a native language speaker to proof their manuscript in term of eliminating the language mistakes and enhance the coherence.
- In addition, the technical check of language, a scientific revision is also needed. For example, abstract should have a sentence about the aim and methodology of the manuscript.
- L28- end the sentence with point.
- Only one reference in 2021 was used, hence the reference needs to be updated and cite the related reference.
- L43, how about the developing worldwide, try to not be too closed with the Chinese preference and make your aim globally.
- The second paragraph of introduction has no aims, try to create why you narrate such literature and connect with the other parts.
- L73, before your make your hypothesis, compare with the available studies around your point to show us the novelty of your study.
- L84… with slight modifications instead.
- L85… space between the unit and degree throughout the text like L113.
- L97… explain the composition of the commercial smoke in detail.
- .. specify the room temperature.
- L136… in details, explain the conditions of HPLC and the gradient, column, etc.
- How you prepared the bacon extract for measuring the antioxidant activity.
- Please provide much clearer and better figures for Figs1, 2, and 3.
- Why was the statistical analysis missed for Shine in Table 1?
- Fig 4, here I have major concern about this analysis, with the values of 52 and 18.2% for PC1 and 2, the output values and correlation are useless, as the PC could be significant and meaningful after reaching the values over 85%. So, either to delete it or repeat it again.
Author Response
Q: So, the novelty of this study is somewhat not clear, and the authors ought to highlight what are the new things they deliver to the scientific community, why we have to read your paper and ignore others, plz consider this point in detail.
R: Thank you for your suggestion, I have revised the introduction (see Introduction in revised manuscript).
Q: L28- end the sentence with point.
R: Revised as suggested (see Line 30 in revised manuscript).
Q: Only one reference in 2021 was used, hence the reference needs to be updated and cite the related reference.
R: Some new references have been added, including reference 1, 4, 10, 11, and 13 in the revised manuscript (marked in red).
Q: L43, how about the developing worldwide, try to not be too closed with the Chinese preference and make your aim globally.
R: The sentence has been revised to make our aim globally (see Line 39-40 in revised manuscript).
Q: Lines 42-44. Not only phenolic compounds but also heat induced compounds, such as Maillard reaction products, might play a role towards coffee health-promoting effects. Implement and add references.
R: More references were added (see Line 37-39 in revised manuscript).
Q: The second paragraph of introduction has no aims, try to create why you narrate such literature and connect with the other parts.
R: The purpose of this study is to develop some liquid smoke with antioxidant property to improve the stability and shelf life of bacon. This paragraph discussed the detrimental effects of lipid and protein oxidation in foods to demonstrate the necessity of our current study. To make our statement clearer, we made some revision and two literatures were added in the paragraph (see Line 53-56 in revised manuscript).
Q: L73, before your make your hypothesis, compare with the available studies around your point to show us the novelty of your study.
R: According to your suggestion, we revised the manuscript with addition of some reference (see Line 53-58 in revised manuscript).
Q: L84… with slight modifications instead.
R: I have revised this sentence as suggested (see Line 69-70 in revised manuscript).
Q: L85… space between the unit and degree throughout the text like L113.
R: Revised as suggested (see Line 70, 83, 113, 120, 138, 155 in revised manuscript).
Q:.. specify the room temperature.
R: The room temperature in this study is 25 °C (see Line 94, 120 in revised manuscript).
Q: L97… explain the composition of the commercial smoke in detail.
R: This is a kind of commercial liquid smoke widely used in meat smoking. Due to technical protection or patent restrictions, we do not have information about the composition of this commercial liquid smoke. Some information about this product was added in the revised manuscript (Line 78-79 in revised manuscript).
Q: L136… in details, explain the conditions of HPLC and the gradient, column, etc.
R: I have added the condition of HPLC analysis (see Line 105-110 in revised manuscript)
Q: How you prepared the bacon extract for measuring the antioxidant activity.
R: I have added the information about how to prepared the bacon extract (see Line 115-117 in revised manuscript)
Q: Please provide much clearer and better figures for Figs1, 2, and 3.
R: I have revised all the figures, and coverted the format of those figures into png (see Figures in revised manuscript).
Q: Why was the statistical analysis missed for Shine in Table 1?
R: We feel sorry, because there is no significant difference in the shine data among different groups, so we forgot to mark the relevant symbols.
Q: Fig 4, here I have major concern about this analysis, with the values of 52 and 18.2% for PC1 and 2, the output values and correlation are useless, as the PC could be significant and meaningful after reaching the values over 85%. So, either to delete it or repeat it again.
R: Principle component analysis (PCA) for sensory attributes of bacon smoked with different woods based on huge amounts of origin data, in this PCA 70.2% of the variance was explained by the first two components (PC1 = 52.0% and PC2 = 18.2%) is acceptable, because in similar studies, it is generally believed that the variance greater than 60% can prove the reliability of the PCA model
Here are the reference:
Zeng, M., Li, Y., He, Z., Qin, F., Tao, G., Zhang, S., & Chen, J. (2016). Discrimination and investigation of inhibitory patterns of flavonoids and phenolic acids on heterocyclic amine formation in chemical model systems by UPLC-MS profiling and chemometrics. European Food Research and Technology, 242(3), 313-319.
Quan, W., Wu, Z., Jiao, Y., Liu, G., Wang, Z., He, Z., & Chen, J. (2021). Exploring the relationship between potato components and Maillard reaction derivative harmful products using multivariate statistical analysis. Food Chemistry, 339, 127853.

Reviewer 2 Report
Thanks for authors, they did a good job. However, there are several points need clarification;
1-Title: requirs modification to be suited with the contents of the article; my recommendation is to changed into (processing of bacon with different type of liquid smoke: Antioxidant capacity). you can change it into other title but please focus the use of smoke on bacon processing
2- The Abstract: the weak part of the study; the first paragraph is of no need; the methodology is ignored at all which is unacceptable in article writting rules; the results id confused; it requirs re arrarrangement
3- Materials and methods: It is very very long; The references and detalis of modification( if present) must be inserted. be carful when writting slight modification because you have to have this modification already; when reviewing the reference you cited I did not find such modification. Please clarify.
4- Why authprs used different methods to measures the total antioxidant capacity. is one method not enough? please clarify.
5- results and discussion: Joining of two parts make me very confused. therefore, I prefer to separate two part for better presentation and evaluation.
Author Response
Q: 1-Title: requirs modification to be suited with the contents of the article; my recommendation is to changed into (processing of bacon with different type of liquid smoke: Antioxidant capacity). you can change it into other title but please focus the use of smoke on bacon processing
R: We have revised the title according to your suggestion (see Line 3 in revised manuscript).
Q: The Abstract: the weak part of the study; the first paragraph is of no need; the methodology is ignored at all which is unacceptable in article writting rules; the results id confused; it requirs re arrarrangement
R: According to your suggestion, I have revised the abstract (see Line 17-20 in revised manuscript).
Q: Materials and methods: It is very very long; The references and detalis of modification( if present) must be inserted. be carful when writting slight modification because you have to have this modification already; when reviewing the reference you cited I did not find such modification. Please clarify.
R: I have revised this section according to your suggestion (see Materials and methods in revised manuscript).
Q: Why authprs used different methods to measures the total antioxidant capacity. is one method not enough? please clarify.
R: Generally speaking, there are many methods to measure and reflect the antioxidant capacity, but different methods have certain limitations. It is not accurate enough to use only one method to determine the antioxidant capacity.
At present, many related literatures use multiple methods to comprehensively evaluate the antioxidant capacity:
de Abreu Pinheiro, F., Elias, L. F., de Jesus Filho, M., Modolo, M. U., Rocha, J. D. C. G., Lemos, M. F., ... & Cardoso, W. S. (2021). Arabica and Conilon coffee flowers: Bioactive compounds and antioxidant capacity under different processes. Food chemistry, 336, 127701.
Schmeda-Hirschmann, G., Antileo-Laurie, J., Theoduloz, C., Jiménez-Aspee, F., Avila, F., Burgos-Edwards, A., & Olate-Olave, V. (2021). Phenolic composition, antioxidant capacity and α-glucosidase inhibitory activity of raw and boiled Chilean Araucaria araucana kernels. Food Chemistry, 350, 129241.
Q: results and discussion: Joining of two parts make me very confused. therefore, I prefer to separate two part for better presentation and evaluation.
R: We originally planned to divide the results and discussion into two parts in the manuscript, but after consideration, the combined form of the existing results and discussion can more clearly reflect the conclusions obtained and the relationship between different results. Futher more, the combined results and discussion complies with the file format of the journal.

Reviewer 3 Report
The reviewed manuscript raises issues related to the use of liquid smoke as a replacement for the traditional method of smoking used so far. The authors point out important issues related to the lower content of harmful substances and the increase in antioxidant activity, which extends the shelf life. The work contains elements of innovation and has a potential practical significance that can be implemented in food technology. I estimate the scientific value as above average. The manuscript is well designed and organized and the experiment itself is properly designed. However, I have comments on the methodological issue and the presentation of the results. Paragrapf 2.1 - preparation of liquid smoke. why the distillation of the wood took only 15 minutes. Is it the time limit associated with unfavorable changes above that time or the maximum efficiency of the process? Line 118, please specify foline-phenol reagent or the same as folin reagent line 121 why methanol was used as a blank or because liquid smoke was dissatisfying, please specify I am also asking you to present the device on which the TPC analyzes were carried out line 136 please provide a detailed protocol for HPLC paragraph 2.4. please describe the device (microplate reader) used for AA activity. figure 2 - E and F charts please change the scale to the same units initially the reader gets the impression that the DPPH and ABTS values ​​are almost identical Figure 3 - A-E charts, in my opinion, presenting the results in the form of line charts would be more legible and would better reflect the dynamics of changes during storage Table 1. in my opinion, replacing the table with a radar chart would be more legible and better represent the sensory properties of bacon
Author Response
Q: Paragrapf 2.1 - preparation of liquid smoke. why the distillation of the wood took only 15 minutes. Is it the time limit associated with unfavorable changes above that time or the maximum efficiency of the process?
R: In our study, only flue gas after 15 min of smoking was collected because (1) it takes a certain time (less than 15 min) for the instrument to heat up to the distillation temperature of 680 °C; (2) the initial flue gas contains a large amount of water vapor in the wood, and the concentration of effective substances related to the smoky flavor is very low. So only the flue gas after 15 min of smoking was collected.
Q: Line 118, please specify foline-phenol reagent or the same as folin reagent
R: The reagent has been specified (see Line 93 in revised manuscript)
Q: line 121 why methanol was used as a blank or because liquid smoke was dissatisfying, please specify I am also asking you to present the device on which the TPC analyzes were carried out
R: I feel sorry about this mistake, distilled water was used as a blank, I have revised this sentence, and the information about the device was added (see Line 96 in revised manuscript).
Q: line 136 please provide a detailed protocol for HPLC
R: I have added the condition of HPLC analysis (see Line 105-110 in revised manuscript)
Q: paragraph 2.4. please describe the device (microplate reader) used for AA activity.
R: The device (microplate reader) used for TPC and AA activity have been added (see Line 94-96 in revised manuscript)
Q: Figure 2 - E and F charts please change the scale to the same units initially the reader gets the impression that the DPPH and ABTS values are almost identical
R: We have revised figure 2E and F according to your suggestion (see Figure 2 in revised manuscript)
Q: Figure 3 - A-E charts, in my opinion, presenting the results in the form of line charts would be more legible and would better reflect the dynamics of changes during storage
R: We tried to present Figure 3 as a line chart, but because the data in Figure 3 contains 7 experimental groups and 4 time points, as well as the standard deviation of these data. These huge amounts of data make it difficult to clearly show the changes and differences of these data in a line chart. After careful consideration, we believe that using a histogram in Figure 3 is more appropriate.
Q: Table 1. in my opinion, replacing the table with a radar chart would be more legible and better represent the sensory properties of bacon
R: According to your suggestion, I have replaced the table with a radar chart (see Figure 1 in revised manuscript)

Round 2
Reviewer 1 Report
The quality of ALL FIGURES still needs to be improved, to be more clear, and understandable.
Author Response
To Referee 1:
Q: The quality of ALL FIGURES still needs to be improved, to be more clear, and understandable.
R: Thank you for your suggestion, I have reproduced these figures, modified some of the annotations in these figures, and adjusted the resolution of the figures to 600dpi to make the figures more clear. In addition, we have made appropriate changes to the figure captions and the serial numbers of the figures quoted in the manuscript. (see Figures in revised manuscript).

Reviewer 2 Report
Accept
Author Response
Thank you for your important revisions to this manuscript
Reviewer 3 Report
All my remmarks have been addressed satisfctory.
Author Response

(The authors gave the same response as above.)
